# Impact of COVID-19 pandemic on utilization of essential maternal healthcare services in Ethiopia: A systematic review and meta-analysis

Birye Dessalegn Mekonnen*, Berhanu Wale Yirdaw

Department of Nursing, Teda Health Science College, Gondar, Ethiopia

* birye22@gmail.com

**Data Availability Statement:** All relevant data are included within the paper.

**Funding:** The authors received no specific funding for this work.

## Abstract

### Background

The COVID-19 pandemic has a significant challenge for countries to maintain the provision of essential maternity services. Many women could experience difficulties in accessing maternal healthcare due to transport problems, anxiety, and fear of infection. A reduction in the utilization of maternity services has been suggested as a possible cause of worsened maternal health outcomes. Thus, this study aimed to determine the impact of the COVID-19 pandemic on the utilization of maternal healthcare services in Ethiopia.

### Methods

Searching of articles was conducted from PubMed, Science Direct, Cochrane Library, Web of Science, Scopus, and Google scholar. The quality of studies was evaluated using the Newcastle-Ottawa scale. Inspection of the Funnel plot and Egger's test were used to evaluate the evidence of publication bias. Heterogeneity was evaluated using Cochran's Q statistic and quantified by $I^2$. A random-effects model was used to determine pooled estimates using STATA 14.

### Results

After reviewing 41,188 articles, 21 studies were included in this systematic review and meta-analysis. The pooled reduction was 26.62% (95% CI: 13.86, 39.37) for family planning, 19.30% (95% CI: 15.85, 22.76) for antenatal care, 12.82% (95% CI: 7.29, 18.34) for institutional delivery, 17.82% (95% CI: 8.32, 27.32) for postnatal care, and 19.39% (95% CI: 11.29, 27.49) for abortion care. This study also demonstrated that maternal perception of poor quality of care and fear of infection, lack of transport, cultural events, diversion of resources, lack of essential drugs, and lack of personal protective equipment and sanitizer were identified as the main challenges faced during the pandemic.

**Competing interests:** The authors have declared that no competing interests exist.

## Conclusion

This study revealed that the utilization of maternal healthcare services in Ethiopia significantly decreased during the COVID-19 pandemic. Government measures, health facility-related barriers, and maternal-related factors were identified as challenges faced during the pandemic. Thus, service providers, policy-makers, and other relevant stakeholders should prioritize maternity care as an essential core healthcare service. Besides, increasing awareness of women through mass media, and making maternity services more accessible and equitable would likely increase the utilization of maternal healthcare services.

## Systematic review registration

PROSPERO CRD42021293681.

## Background

The COVID-19 pandemic has detrimental effects on global healthcare systems, the world economy, and societal structures [1, 2]. Nationwide lockdowns, fear of attending healthcare facilities, and disruption of healthcare services caused by the COVID -19 pandemic could have affected the well-being of mothers and their babies [3, 4]. Maternal and child health services are facing a challenge generated by the COVID-19 pandemic [4–7].

A study conducted to estimate the possible effect of the COVID-19 pandemic on the utilization of sexual and reproductive health in low and middle-income countries indicated that a modest decline of 10% in coverage of essential maternal and newborn healthcare services was reported due to coronavirus pandemics which would result in 28,000 maternal deaths [8].

The COVID-19 pandemic disrupts healthcare services which further leads to a reduction of essential maternal and child health interventions in low and middle-income countries [3, 9, 10]. A reduction in the provision of maternity services, and changed healthcare-seeking behavior have been suggested as a possible cause of worsened maternal health outcomes [11, 12]. A systematic review and meta-analysis revealed an increase in maternal deaths, ectopic pregnancies, maternal depression, ruptured, and stillbirth [13]. Literature indicates that the risk of maternal intensive care unit admission and maternal mortality have increased during the COVID-19 pandemic [14]. Emerging evidence also suggests that fetal outcomes have worsened due to the COVID-19 pandemic with an increase in rates of preterm birth and stillbirth [15, 16].

The COVID-19 pandemic has a significant challenge for many countries to maintain the provision of essential maternal, newborn, and child health services [3, 17, 18]. Many women could experience difficulties in accessing maternal healthcare due to transport problems, restrictions, anxiety, and fear of probably being exposed to coronavirus disease [19–22]. A systematic review and meta-analysis has been demonstrated a significant decrease in the utilization of essential maternal healthcare services [23].

The diversion of resources away from basic maternal, newborn, and child health services, because of prioritizing the coronavirus pandemic response leads to the disruption of maternal health services and increased risks of maternal illness and death [24]. This could further be a challenge for providing essential services along the maternity continuum of care while battling with COVID-19 [25]. Recent evidence indicated that government measures against COVID-19 such as stay at home guidance, women's healthcare-seeking behavior, community

perception, perceived poor quality of care during the pandemic, and fear of contracting COVID-19 were challenges that influence maternity care provided to mothers during pregnancy, childbirth, and the postpartum period [13, 22, 26–28].

Ethiopia reported the first confirmed case of COVID-19 on March 13, 2020 [29]. In Ethiopia, following the declaration of the COVID-19 pandemic, local mobility was restricted, gatherings in all settings were not allowed, and individuals who suspect of having acquired the virus should report to the nearby health authority [30, 31]. Similarly, international travelers were not permitted to board a flight for entry, or transit without a negative COVID-19 test result, while those who entered are expected to stay home for 14 days before mixing with others [30, 31].

In Ethiopia, except for fragmented studies with varying reports, there is no nationwide evidence that indicates the effect of COVID-19 on essential maternal healthcare services utilization. Also, government and health facility-related barriers, and individual and community perception-related challenges that possibly influence the utilization of maternal healthcare services amid COVID-19 have not been well described. Thus, this systematic review and meta-analysis aimed to fill this gap by estimating the pooled reduction and challenges of essential maternal healthcare services utilization in Ethiopia.

## Methods

This systematic review and meta-analysis was conducted and reported following the Preferred Reporting Items for Systematic Reviews and Meta-Analyses (PRISMA) checklist [32] (S1 Table). The systematic review was prospectively registered on the International Prospective Register of Systematic Reviews (PROSPERO) with the unique number CRD42021293681.

### Inclusion criteria

This review follows the population, intervention, comparator, and outcome (PICO) framework with (P) women of reproductive age, or/and female adolescents, (I) utilization of essential maternal health services, (C) no use of essential maternal health services, (O) effect of Covid-19 on essential maternal health services. Accordingly, studies that fulfill the following inclusion criteria were included: (1) Quantitative or qualitative or mixed methods studies that reported data on the change in the utilization of maternal health services (family planning, antenatal care, institutional delivery, postnatal and abortion care) in Ethiopia were considered; (2) Both peer-reviewed published and unpublished articles (pre-prints and grey literature) were considered; (3) Articles written and published in the English language from December 2019 to November 20, 2021, were included.

### Exclusion criteria

Studies that described service adaptation and mitigation strategies, and did not include data on maternal health services utilization were excluded. Besides, letters, case reports and series, editorial reports, commentaries, reviews, and guidelines were excluded from the study.

### Search strategy and information sources

A comprehensive search of literature from PubMed/Medline, Science Direct, Cochrane Library, Web of Science, Scopus, and Google scholar was conducted from October 21, 2021, to November 20, 2021. The search was done using Medical Subject Headings (MeSH) with the following search terms: 'impact', 'effect' 'influence', 'COVID-19', 'SARS-CoV-2', 'coronavirus', 'novel coronavirus', 'coronavirus disease 2019', 'maternal health service', 'maternal care

service', 'health service utilization', 'health care utilization, 'family planning service', 'family planning use', 'antenatal care', 'prenatal care', 'skilled birth attendant', 'institutional delivery', 'health facility delivery', 'postnatal care', 'postpartum care', 'abortion care', abortion service', and 'Ethiopia'. A combination of different Boolean operators (AND, OR), and truncation were used to develop the search strategies. Likewise, the reference lists of all the studies identified as relevant were also manually reviewed to catch studies that were not captured by electronic articles searches.

## Study selection

All the retrieved articles were exported to EndNote X7 reference manager software to manage duplicate studies, and the screening process [33]. Two reviewers (BDM and BWY) independently assessed the titles and abstracts of remained articles after removing the duplicate articles from EndNote Library. The full texts of studies were also reviewed to confirm their eligibility according to the preset criteria. Differences between the two reviewers were resolved by discussion. The overall study selection screening and processes were summarized using the PRISMA flow diagram.

## Risk of bias (quality) assessment

The Newcastle-Ottawa scale (NOS) tool adapted for cross-sectional studies quality assessments was used to assess the quality of each study. Two reviewers (BDM and BWY) independently assessed the methodological quality of all articles selected in the review. Any discrepancy between the two authors was resolved through discussion. The assessment instrument contains 10 points (stars) in three main sections. The first part of the tool weighted as five points focuses on the selection which is the methodological quality of each study. The second part of the tool focused on the comparability of the study which rated two points. The last part is focused on the assessment of the outcomes and statistical tests of the primary study with a possibility of three points. Finally, original studies assessed with a score of $\geq 7$ out of 10, 5–7 out of 10, and $\leq 4$ out of 10 were considered as achieving high, medium, and low quality respectively (S2 Table).

## Data extraction

A standardized data extraction tool, which was adapted from the Joanna Briggs Institute (JBI) was used to extract data from articles included in the review [34]. Two authors (BDM and BWY) independently extracted the data after a screening of titles, abstracts, and the full texts of each primary study included in this meta-analysis. Any variance between the two authors was resolved by consensus after discussion. The following necessary information was extracted from each included article: the name of the author, study region and setting, year of publication, study design, study participants, sample size, percentage of change in family planning, antenatal care, institutional delivery, postnatal and abortion care service utilization, and challenges of maternal health services utilization amid of COVID-19 pandemic.

## Data synthesis and analysis

The data were extracted using Microsoft Excel datasheet from included studies and then imported into STATA version 14 meta-analysis. A meta-analysis and qualitative synthesis of the evidence were performed. The qualitative data was analyzed based on the main challenges that could decrease the utilization of essential maternal health services during the COVID-19 pandemic. The findings of selected studies were summarized using tables and figures, and a

forest plot. Heterogeneity among the included studies was evaluated using Cochran's Q statistic and quantified by $I^2$ statistics, and presented by forest plot. The presence of heterogeneity across the selected articles was considered when $p < 0.1$ or $I^2 > 50\%$ [35]. A random-effects model was used to determine pooled estimates as considerable heterogeneity was exhibited between selected articles. To address variations in the primary studies, subgroup analysis was carried out based on publication year (2020 vs 2021).

Visual inspection of funnel plots asymmetry and Egger's regression test were used to evaluate the evidence of publication bias. Egger's test regression was assumed indicative of publication bias if a p-value < 0.05 [36]. Nonparametric trim and fill analysis was conducted using the random-effect analysis in case evidence of publication bias was observed [37]. Tests of publication bias for postnatal and abortion care were not executed as they could be highly underpowered because of the limited number of included studies. Besides, a sensitivity analysis was conducted to check the presence of a single study that influence the overall estimate [38].

## Results

### Study selection

Initially, a total of 41,188 potentially relevant articles were yielded through search strategy. After removing 8,761 duplicates, 32,427 unique studies remained. Two authors (BDM and BWY) screened the remaining articles based on their titles and abstracts which result in the exclusion of 32,315 articles. Then 112 full-text articles were evaluated for eligibility based on the inclusion and exclusion criteria. Of the 112 articles, 91 were excluded due to variation in study locations, study population, no relevant data, commentary, and duplicate. Lastly, 21 studies were included in the final systematic review and meta-analysis (Fig 1).

### Study characteristics

Out of twenty-one selected studies; thirteen studies were quantitative cross-sectional studies, five were mixed (quantitative and qualitative) studies, two were pre-post studies, and one was a qualitative study. Sixteen studies were published [39–54] while the other five studies were unpublished [55–59]. Seven studies were conducted in Addis Ababa administrative city [39, 42, 43, 46, 47, 55, 58], three studies were conducted in Southern Nations Nationalities and People (SNNPR) [44, 51, 59], four studies were from Amhara region [40, 45, 49, 57], four studies were nationwide surveys [48, 50, 52, 56], one study was in Oromia region [53], one study was from Tigray region [54], and one study was conducted in Somali region [41]. Eleven studies reported quantitative findings on the reduction of antenatal care, ten studies reported the reduction of family planning, six studies reported a decrease in facility delivery, four studies reported the reduction of abortion care, and three studies reported a reduction in postnatal care services. All the primary studies scored $\geq 6$ out of 10 on the quality assessment point (Table 1).

### Sensitivity analysis

Sensitivity analysis was conducted using the random-effects model for the service reduction estimates of family planning, antenatal care, institutional delivery, postnatal care, and abortion care amid COVID-19. The results of the sensitivity analysis suggested that there is no influential study as none of the points estimate outside of the overall 95% confidence interval for all estimates.

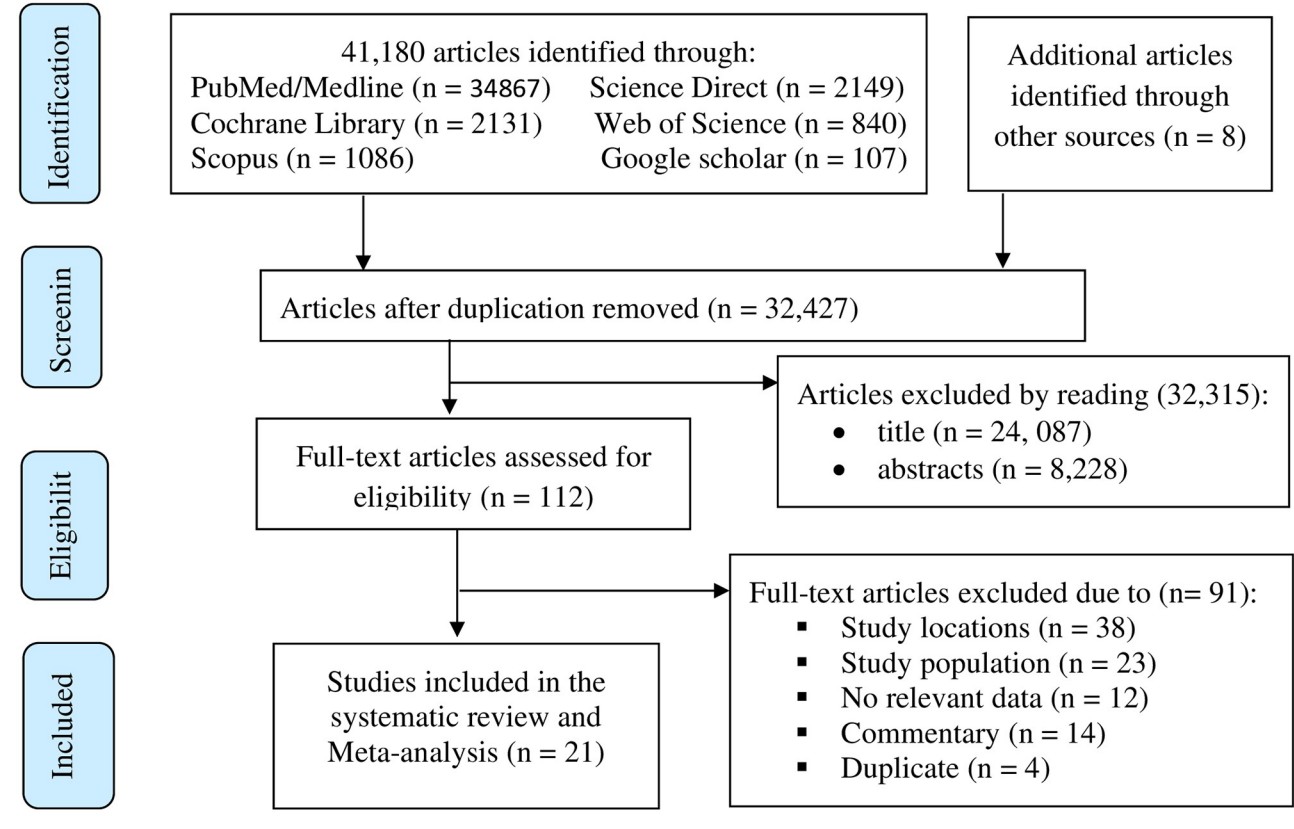

**Fig 1. PRISMA flow diagram of study selection.**

## Impact of COVID-19 pandemic on utilization of maternal healthcare services

### Impact of COVID-19 pandemic on family planning services

Ten studies reported that utilization of family planning services significantly decreased during the COVID-19 pandemic. The pooled proportion of family planning services utilization in Ethiopia was decreased by 26.62% (95% CI: 13.86, 39.37). We used a random-effects model due to the presence of high heterogeneity ($I^2$ = 100%; p = 0.001) in included studies (Fig 2).

Subgroup analysis was carried out based on publication year (2020 vs 2021). Accordingly, the highest (54.46%) family planning services reduction was observed in 2020, while the lowest (16.59%) family planning services reduction was observed in 2021 (Fig 3).

The result of Egger's test (p = 0.248) and the symmetric distribution of studies in the funnel plot declared the absence of publication bias (Fig 4).

### Impact of COVID-19 pandemic on ANC services

According to the report of eleven studies, utilization of antenatal care services considerably decreased during the COVID-19 pandemic. The pooled proportion of antenatal care services utilization in Ethiopia was decreased by 19.30% (95% CI: 15.85, 22.76). We used random effects due to the presence of high heterogeneity ($I^2$ = 99.3%; p = 0.000) in included studies (Fig 5).

**Table 1. Description of studies on the impact of the COVID-19 pandemic on the utilization of maternal healthcare services in Ethiopia, 2021.**

| Author | Year | Region | Study area | Study design | Study Population | Key findings | | | | | NOS score |
|---|---|---|---|---|---|---|---|---|---|---|---|
| | | | | | | FP | ANC | Facility delivery | PNC | Abortion care | |
| Ayele et al. [39] | 2021 | Addis Ababa | Addis Ababa | CS | Pregnant women | | Reduced by 1.3% | | | | 9 |
| Shimels T. [42] | 2021 | Addis Ababa | Addis Ababa | Mixed study | Reproductive age women | Decreased | Decreased | Decreased | Decreased | Decreased | 8 |
| Dandena et al. [43] | 2021 | Addis Ababa | SPHMMC | CS | Reproductive age women | Reduced by 47% | Reduced by 15% | Reduced by 9% | Reduced by 16% | | 8 |
| Kassie et al. [44] | 2021 | SNNPR | Southern Ethiopia | CS | Reproductive age women | Reduced by 15.9% | Reduced by 27.4% | Reduced by 23.5% | Reduced by 29.1% | | 9 |
| Abdela et al. [45] | 2020 | Amhara | Northeast Ethiopia | CS | Reproductive age women | Reduced by 95% | Reduced by 50% | | | | 7 |
| Tolu et al. [46] | 2020 | Addis Ababa | SPHMMC | CS | Reproductive age women | Reduced by 27% | | Reduced by 27.6% | | Reduced by 20.3% | 7 |
| Gebreegziabher et al. [55] | 2021 | Addis Ababa | Addis Ababa | CS | Reproductive age women | Reduced by 20.3% | Reduced by 7% | Reduced by 2.5% | Reduced by 9.3% | Reduced by 23.7% | 8 |
| Bantalem et al. [47] | 2020 | Addis Ababa | Addis Ababa | Mixed study | Reproductive age women | | Reduced by 12% | | | | 6 |
| Shuka et al. [56] | 2021 | Nationwide | Nationwide | CS | Reproductive age women | Reduced by 16% | Reduced by 17% | | | Reduced by 25% | 8 |
| Desta et al. [54] | 2021 | Tigray | Northern Ethiopia | Pre-post study | Reproductive age women | Reduced by 4.81% | Reduced by 2.83% | | | Reduced by 12.3% | 9 |
| Oladeji et al. [41] | 2020 | Somali | Farfan zone | CS | Reproductive age women | | Reduced by 14.4% | Reduced by 21.4% | | | 6 |
| Tadesse E. [40] | 2020 | Amhara | Northeast Ethiopia | CS | Pregnant women | | Reduced by 55.5% | | | | 9 |
| Temesgen et al. [57] | 2020 | Amhara | Northeast Ethiopia | Mixed study | Reproductive age women | Reduced by 60% | Reduced by 32.7% | | | | 9 |
| Seme A et al. [48] | 2021 | Nationwide | Nationwide | CS | Adolescent women | Reduced by 3.5% | | Reduced by 3.5% | | | 6 |
| Enbiale et al. [49] | 2021 | Amhara | Northern Ethiopia | CS | Reproductive age women | Reduced by 14% | | | | | 8 |
| UNICEF [58] | 2020 | Addis Ababa | Addis Ababa | CS | Women and children | | | Reduced by 4.3% | | | 7 |
| Workicho et al. [50] | 2021 | Nationwide | Nationwide | Mixed study | Reproductive age women | | Decreased | Decreased | | | 8 |
| Hailemariam et al. [51] | 2021 | SNNPR | Southern Ethiopia | Qualitative study | Pregnant women | | Decreased | | | | NA |
| Zimmerman et al. [52] | 2021 | Nationwide | Nationwide | Pre-post study | Pregnant and postpartum women | | | Decreased | | | 8 |
| Temesgen et al. [53] | 2021 | Oromia | Central Ethiopia | CS | Pregnant and postpartum women | Decreased | Decreased | Decreased | Decreased | Decreased | 9 |
| Tilahun et al. [59] | 2021 | SNNPR | Southwest Ethiopia | Mixed study | Women who gave birth | | | Decreased | | | 7 |

ANC: Antenatal care, CS: Cross-sectional, FP: Family planning, PNC: Postnatal care, NA: not assessed

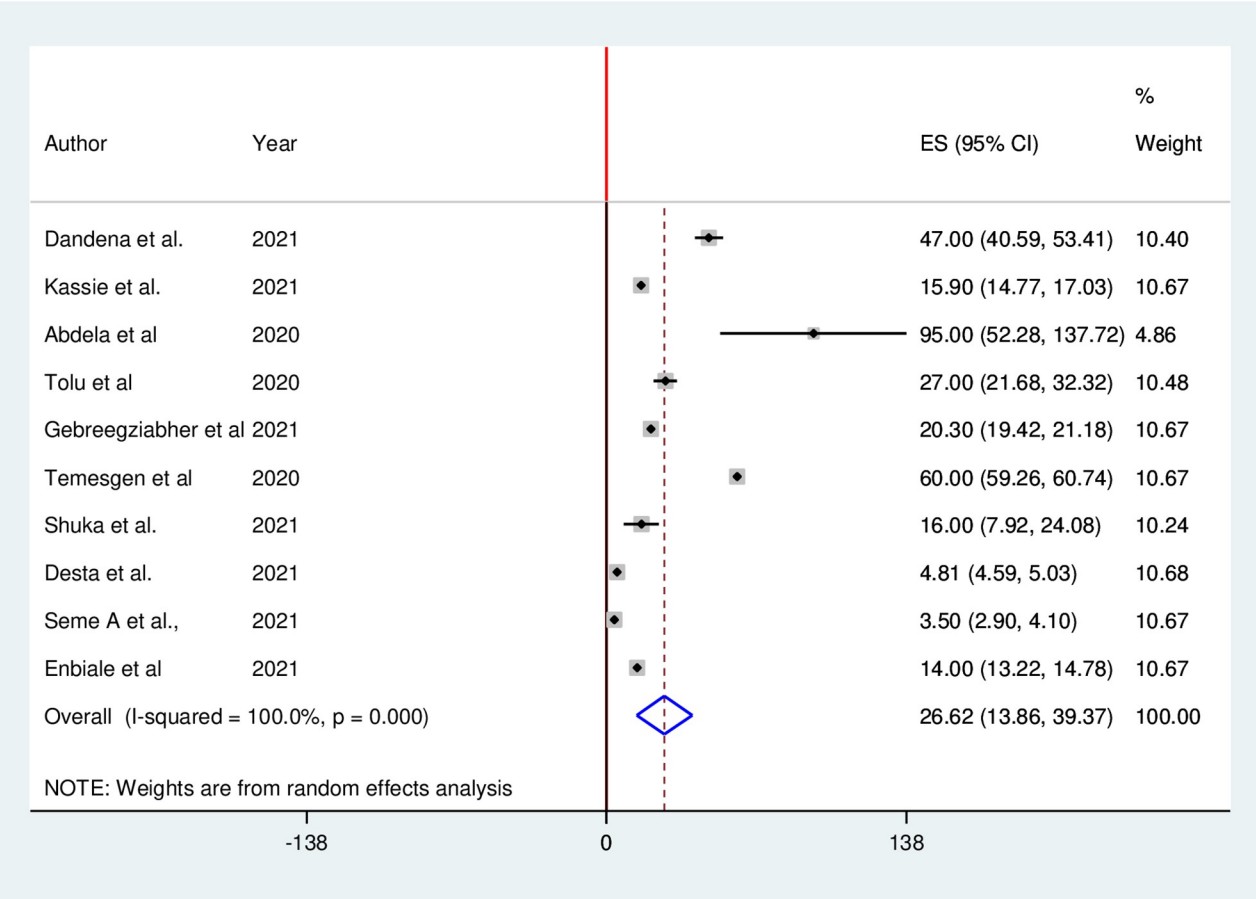

**Fig 2. Forest plot of the pooled reduction of family planning services utilization during COVID-19 pandemic in Ethiopia, 2021.**

Subgroup analysis was carried out based on publication year (2020 vs 2021). Accordingly, the highest (32.49%) antenatal care services reduction was observed in 2020, while the lowest (10.99%) antenatal care services reduction was observed in 2021 (Fig 6).

The result of Egger's test (p = 0.001) and the asymmetric distribution of studies in the funnel plot declared the presence of publication bias. As a result, nonparametric trim and fill analysis was conducted using the random-effect analysis (Fig 7).

## Impact of COVID-19 pandemic on institutional delivery services

Seven studies reported that utilization of institutional delivery services significantly decreased during the COVID-19 pandemic. The pooled proportion of institutional delivery services utilization in Ethiopia was decreased by 12.82% (95% CI: 7.29, 18.34). We used random effects due to the presence of high heterogeneity ($I^2$ = 98.4%; p = 0.000) in included studies (Fig 8).

Subgroup analysis was carried out based on publication year (2020 vs 2021). Accordingly, the highest (17.67%) institutional delivery services reduction was observed in 2020, while the lowest (9.43%) institutional delivery services reduction was observed in 2021 (Fig 9).

The result of Egger's test (p = 0.014) and the asymmetric distribution of studies in the funnel plot declared the presence of publication bias. As a result, nonparametric trim and fill analysis was conducted using the random-effect analysis (Fig 10).

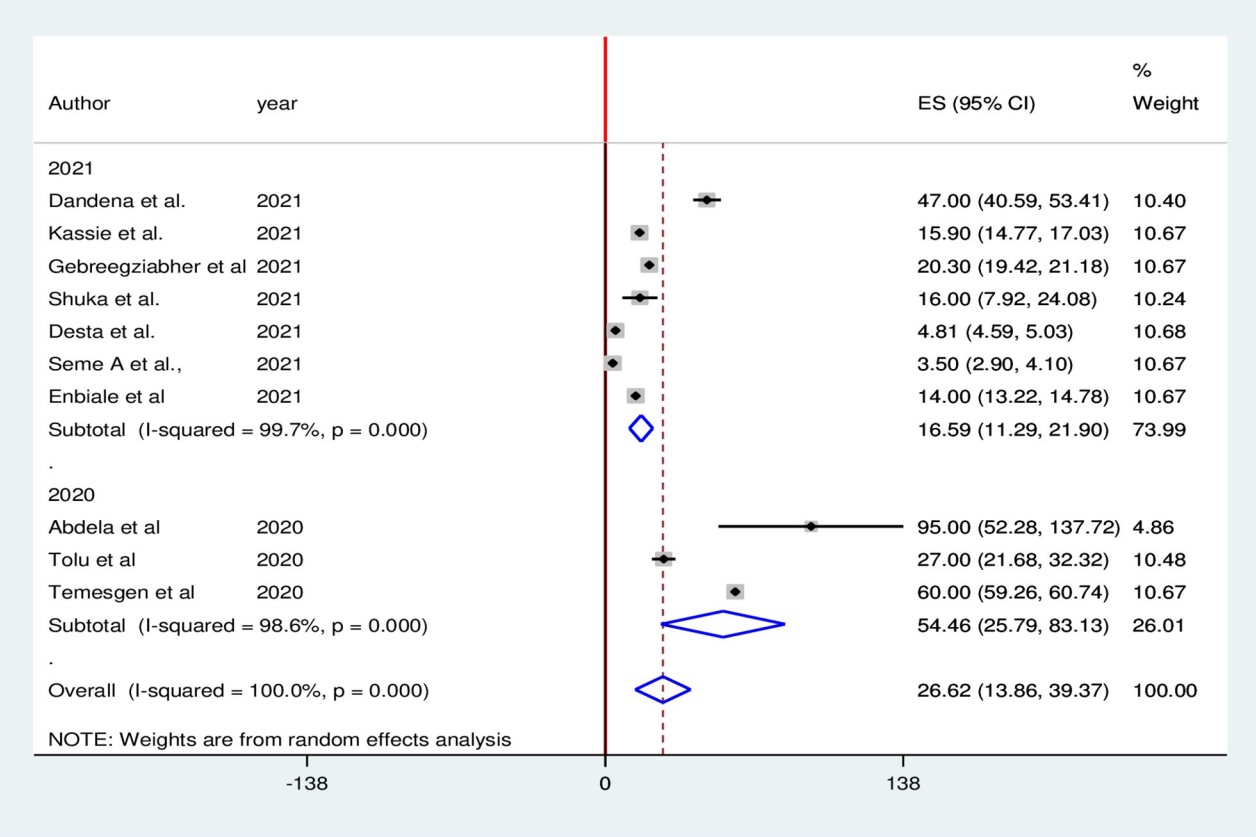

**Fig 3. Subgroup analysis of family planning services reduction during the COVID-19 pandemic in Ethiopia, 2021.**

## Impact of COVID-19 pandemic on postnatal care services

Three studies reported that utilization of postnatal care services significantly decreased during the COVID-19 pandemic. The pooled proportion of postnatal care services utilization in Ethiopia was reduced by 17.82% (95% CI: 8.32, 27.32). We used random effects due to the presence of high heterogeneity ($I^2$ = 97.3%; p = 0.000) in included studies (Fig 11).

## Impact of the COVID-19 pandemic on abortion care services

Four studies reported that utilization of abortion care services significantly decreased during the COVID-19 pandemic. The pooled proportion of abortion care services utilization in Ethiopia was reduced by 19.39% (95% CI: 11.29, 27.49). We used random effects due to the presence of high heterogeneity ($I^2$ = 97.7%; p = 0.000) in included studies (Fig 12).

## Challenges of maternal healthcare services utilization

Ten studies have reported the main challenges that could decrease the utilization of essential maternal health services during the COVID-19 pandemic. These studies showed that there was a significant reduction in the utilization of family planning, antenatal care, institutional delivery, postnatal care, and abortion care. We categorized the challenges into government measures, health facility-related barriers, and maternal-related factors.

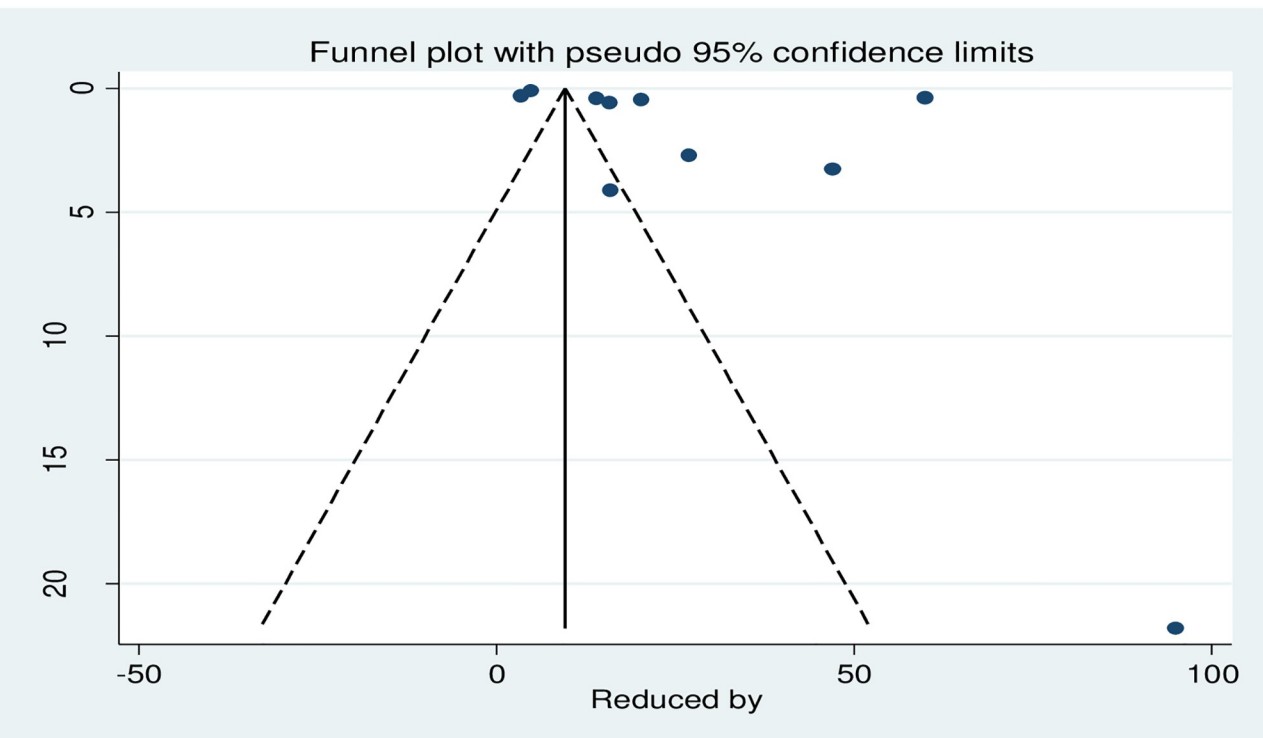

**Fig 4. Funnel plot of all included studies.**

Five studies [40, 47, 51, 53, 57] demonstrated that government measures such as diversion of resources, nationwide lockdowns and restrictions were identified barriers that affected the utilization of maternal healthcare services during the COVID-19 pandemic. Five studies [42, 50, 51, 53, 57] also documented that health facility-related barriers such as staff workload and burnout, lack of essential drugs, mistreatment and disrespect, weak infrastructure, lack of personal protective equipment and sanitizer were the main challenges for utilization of maternal healthcare services during the COVID-19 pandemic. Ten studies [40, 42, 45, 47, 50–53, 57, 59] showed that maternal-related factors such as maternal perception of poor quality of care, lack of transport, cultural events, low economic status, and anxiety and fear of infection and stigma of COVID-19 were explained as the main barriers that affected the utilization of essential maternal health services during the pandemic.

## Discussion

This systematic review and meta-analysis summarized the available national data on the effects of the COVID-19 pandemic on maternal healthcare services utilization in Ethiopia. The study found that almost all of the essential maternal healthcare services significantly decreased during the COVID-19 pandemic. Similar findings have been documented in other countries which reported the impacts of the COVID-19 pandemic on basic maternal and child healthcare services [60, 61]. This considerable decrease could be explained by the inability of healthcare systems to handle the pandemic, reallocation of resources to battle the COVID-19 pandemic, and women could face difficulties in trying to access healthcare services or women could intentionally miss their visits due to fear of contracting the disease [19, 62]. This could also be due to deploying healthcare workers in public health or general medical areas during

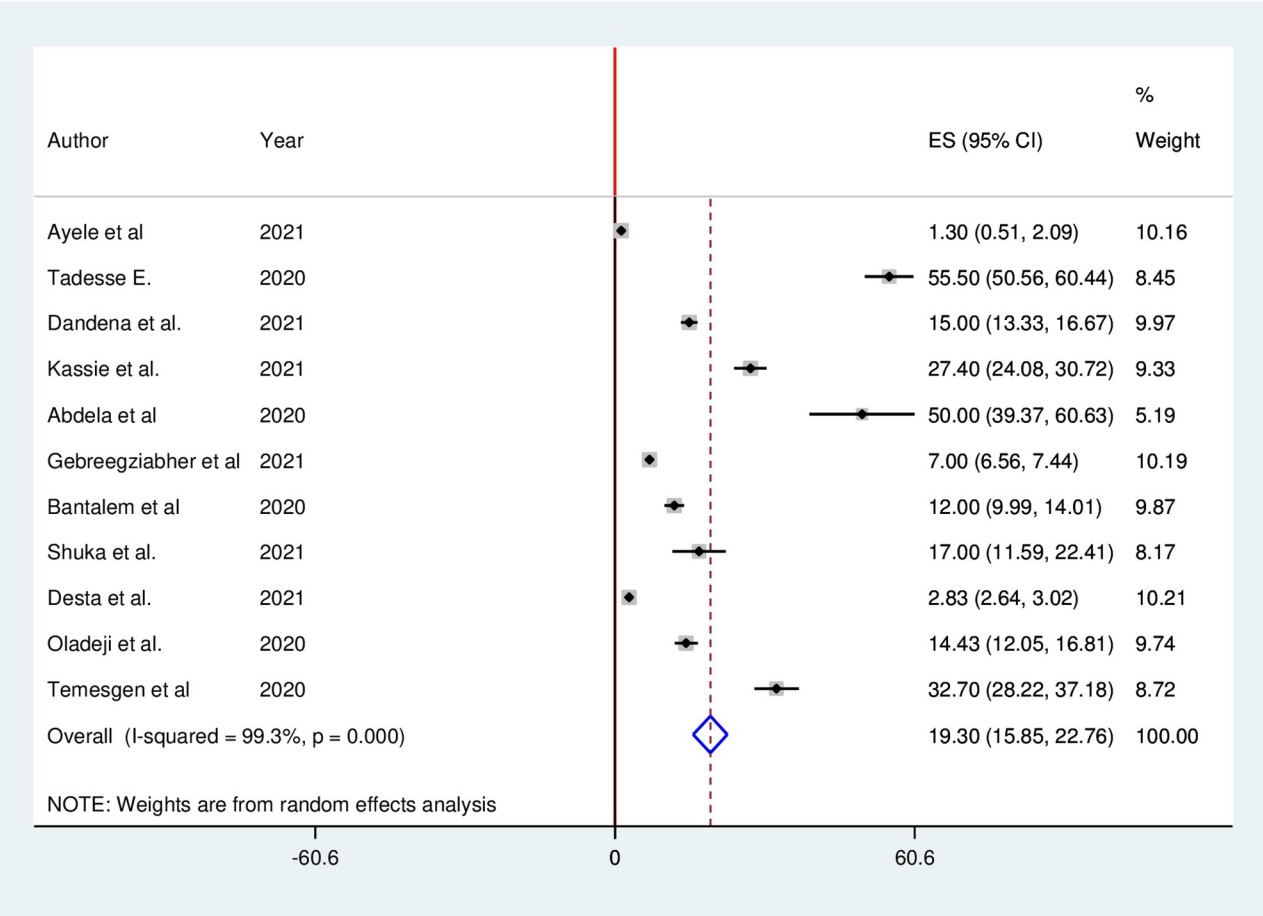

**Fig 5. Forest plot of the pooled reduction of antenatal care services utilization during COVID-19 pandemic in Ethiopia, 2021.**

this pandemic is likely to decrease maternal and newborn health care services [63]. Moreover, the decline could be energized by pre-existing challenges such as poor road conditions and infrastructures, long waiting times, and poor quality of services [3, 64].

This meta-analysis revealed that the pooled proportion of family planning services utilization in Ethiopia decreased by 26.62%. A similar reduction in family planning was reported in India (36%) [65], Italy (16.1%) [66], and Turkey (24.1%) [67]. This could be attributed to limited access to family planning services generated by the COVID-19 pandemic. This highlights the need for continued support from governments, and other concerned bodies to promote family planning services, in line with the Global family planning services collective's call to action during COVID-19. In addition, both the maternity care provider and women should be alert to the service provision.

In this study, the pooled proportion of antenatal care services utilization in Ethiopia decreased by 19.30%. This finding is higher than a 12% of antenatal care reduction in the United Kingdom during the COVID-19 lockdown [68]. However, this finding is lower than the 53% decrease in antenatal care services in Belgium [69]. The findings of both the current and previous studies indicated that the Covid-19 pandemic influenced pregnant women's current pregnancy follow-up to some extent. Furthermore, these findings could be attributed to the fact that the impact of COVID-19 is worldwide [70, 71]. However, the variations could be

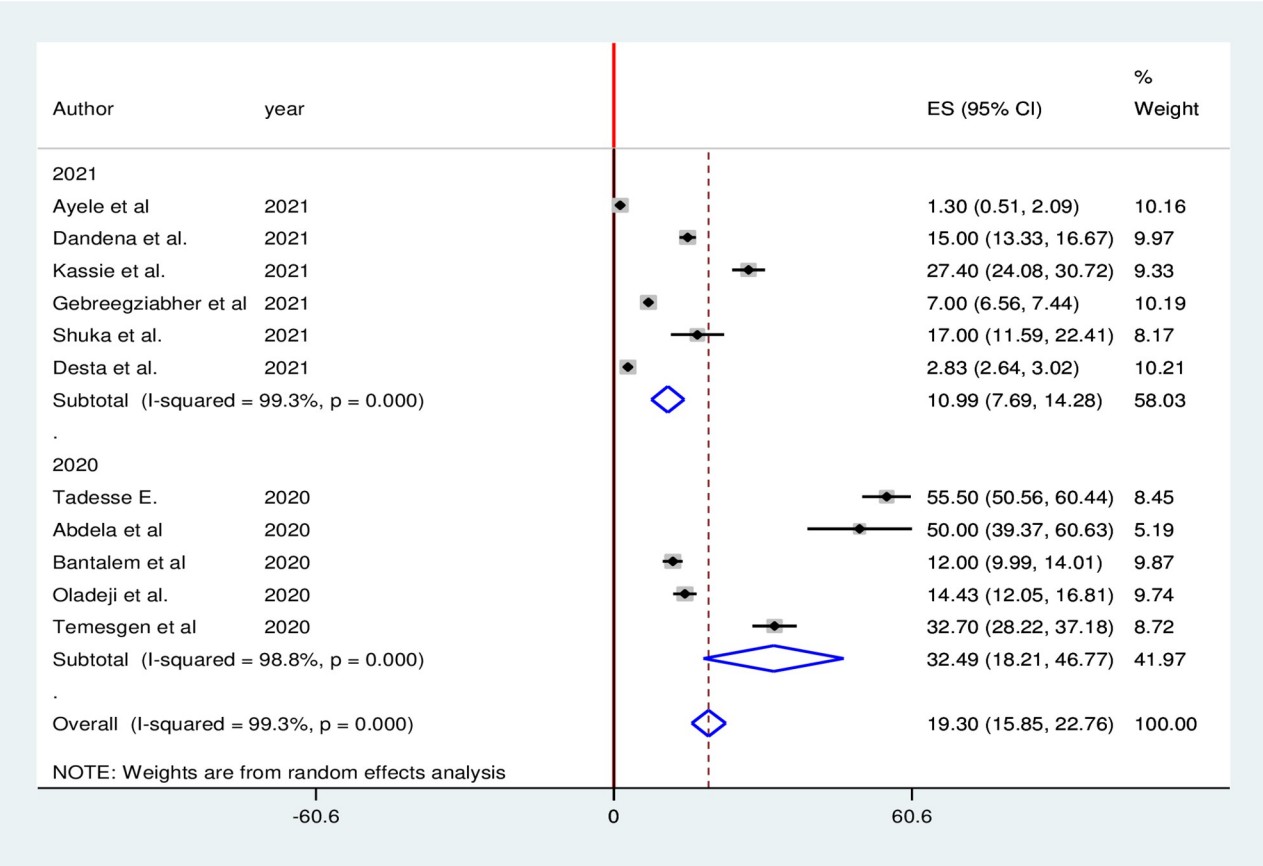

**Fig 6. Subgroup analysis of antenatal care services reduction during the COVID-19 pandemic in Ethiopia, 2021.**

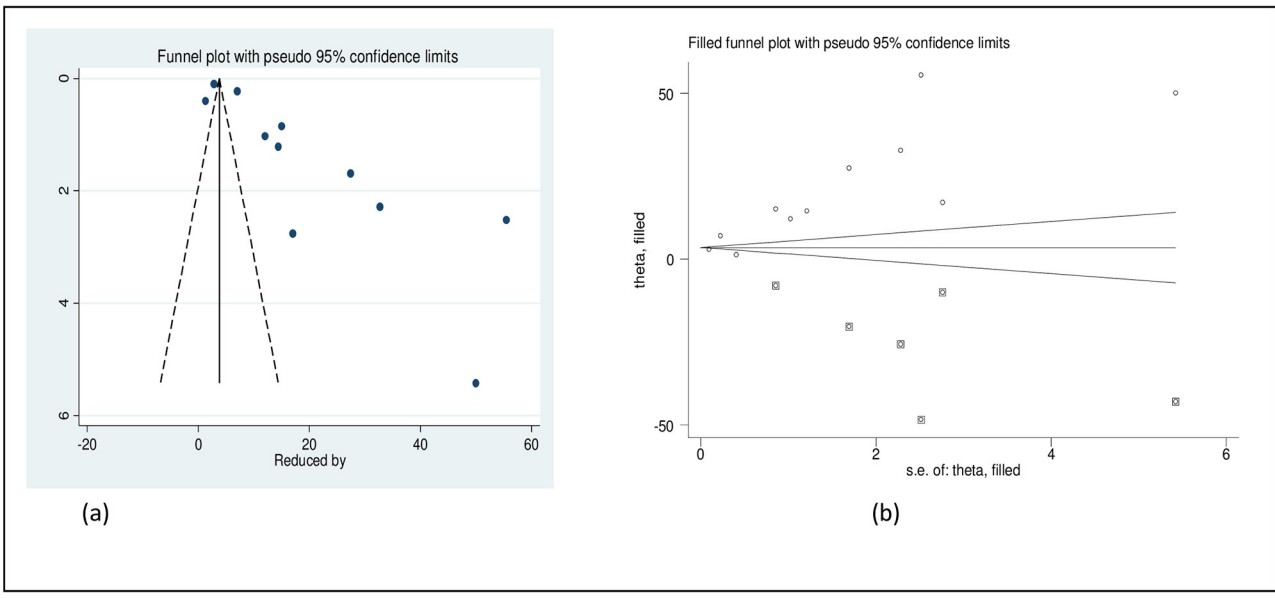

**Fig 7. Original funnel plot (a) and filled funnel plot (b) of all included studies.**

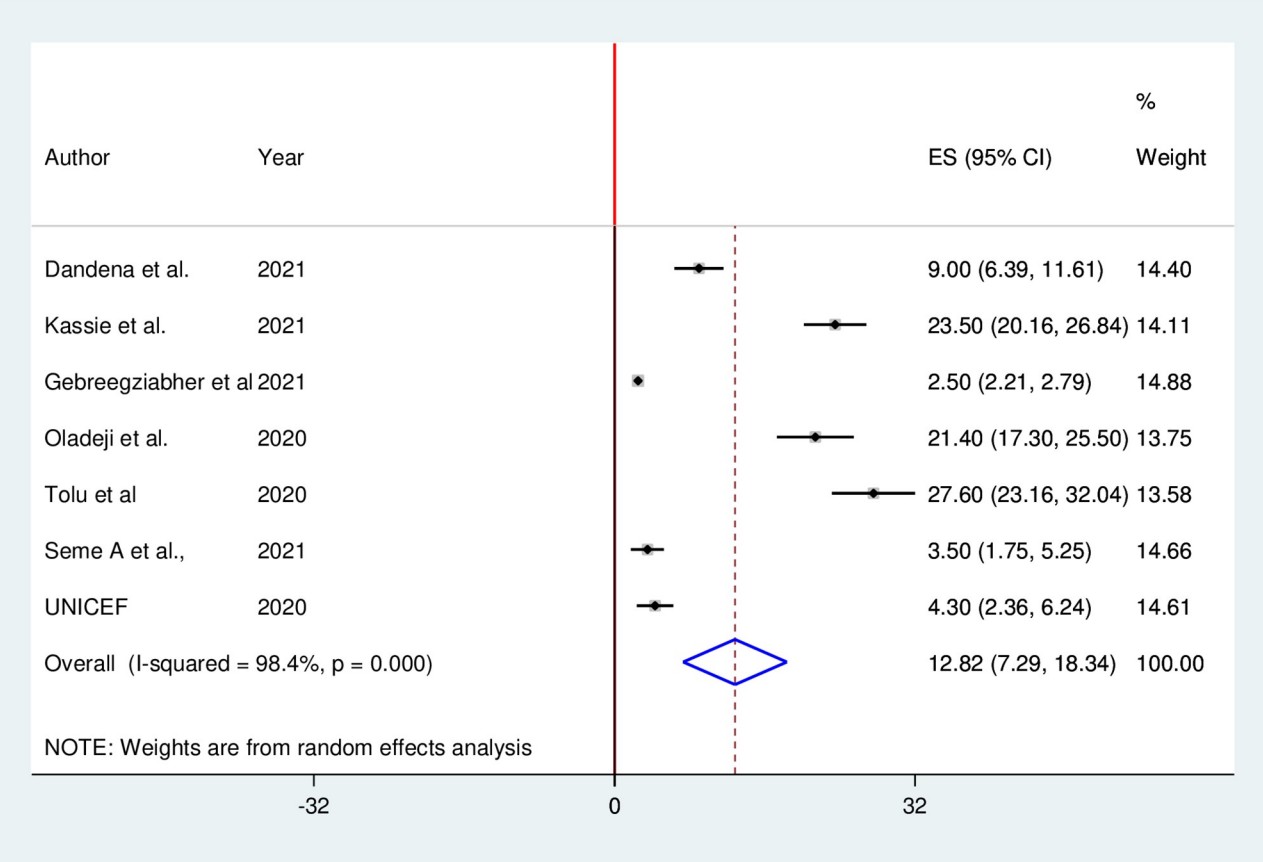

**Fig 8. Forest plot of the pooled reduction of institutional delivery services utilization during COVID-19 pandemic in Ethiopia, 2021.**

due to differences in participants' local norms and cultures, awareness of the severity of the disease, and exposure to credible social media disseminating factual information regarding COVID-19, the burden of the COVID-19 pandemic across the countries.

The results of this systematic review and meta-analysis indicated that the pooled proportion of institutional delivery services utilization in Ethiopia was decreased by 12.82%. A significant reduction in institutional delivery during COVID-19 was also reported in a study done in Nepal [72]. The possible reason could be that pregnant women and community members might not consider home delivery riskier than the COVID-19 pandemic. During the COVID-19 pandemic, a reduction in institutional delivery services utilization would plausibly occur due to disruptions to the enabling environment, and limitations in the use and provision of maternity health services [73, 74]. Furthermore, increased reluctance by women to use the health system and limitations in the availability of skilled health workers could lead to a decrease in institutional delivery service utilization.

The findings of this study indicated that the pooled proportion of postnatal care services utilization in Ethiopia was reduced by 17.82%. This finding was consistent with a study finding done in Nepal which reported a substantial reduction in postnatal care service during COVID-19 [72]. This could be explained by the fact that women may visit health facilities for postnatal care services if they faced complications.

The results of this study indicated that the pooled proportion of abortion care services utilization in Ethiopia was reduced by 19.39%. This finding was supported by findings of previous

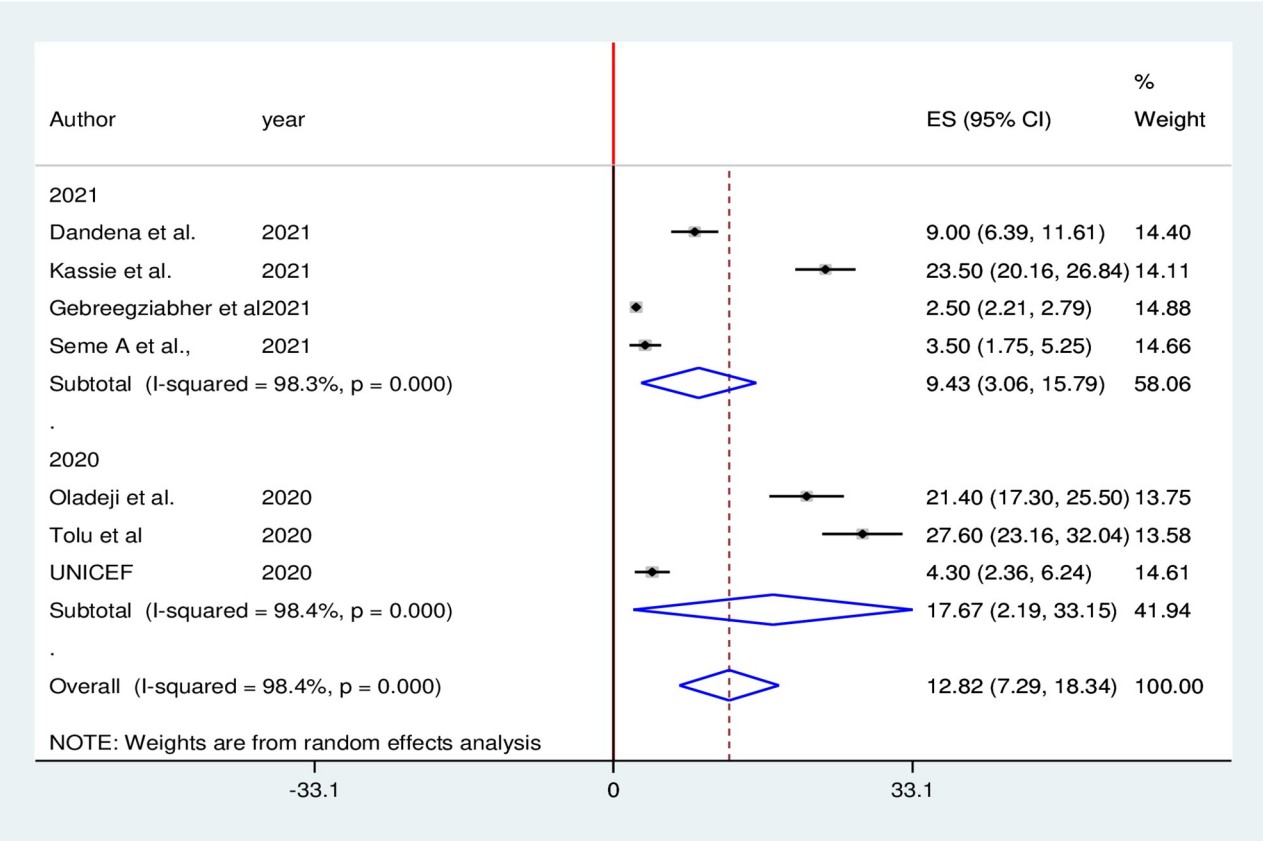

**Fig 9. Subgroup analysis of institutional delivery services reduction during COVID-19 pandemic in Ethiopia, 2021.**

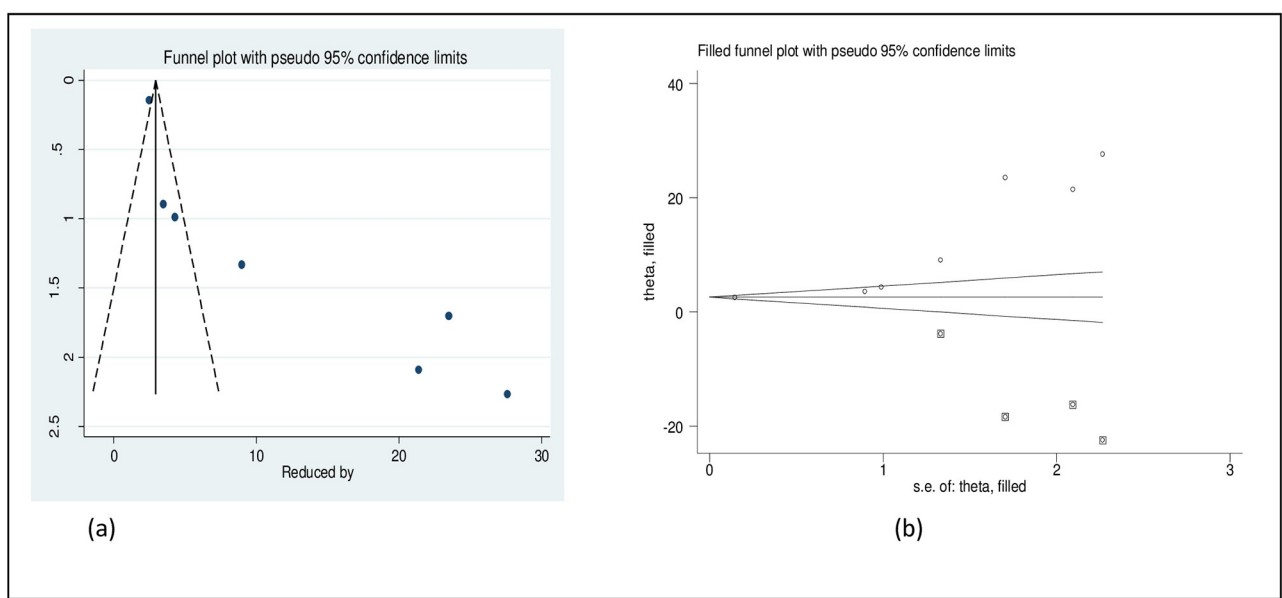

**Fig 10. Original funnel plot (a) and filled funnel plot (b) of all included studies.**

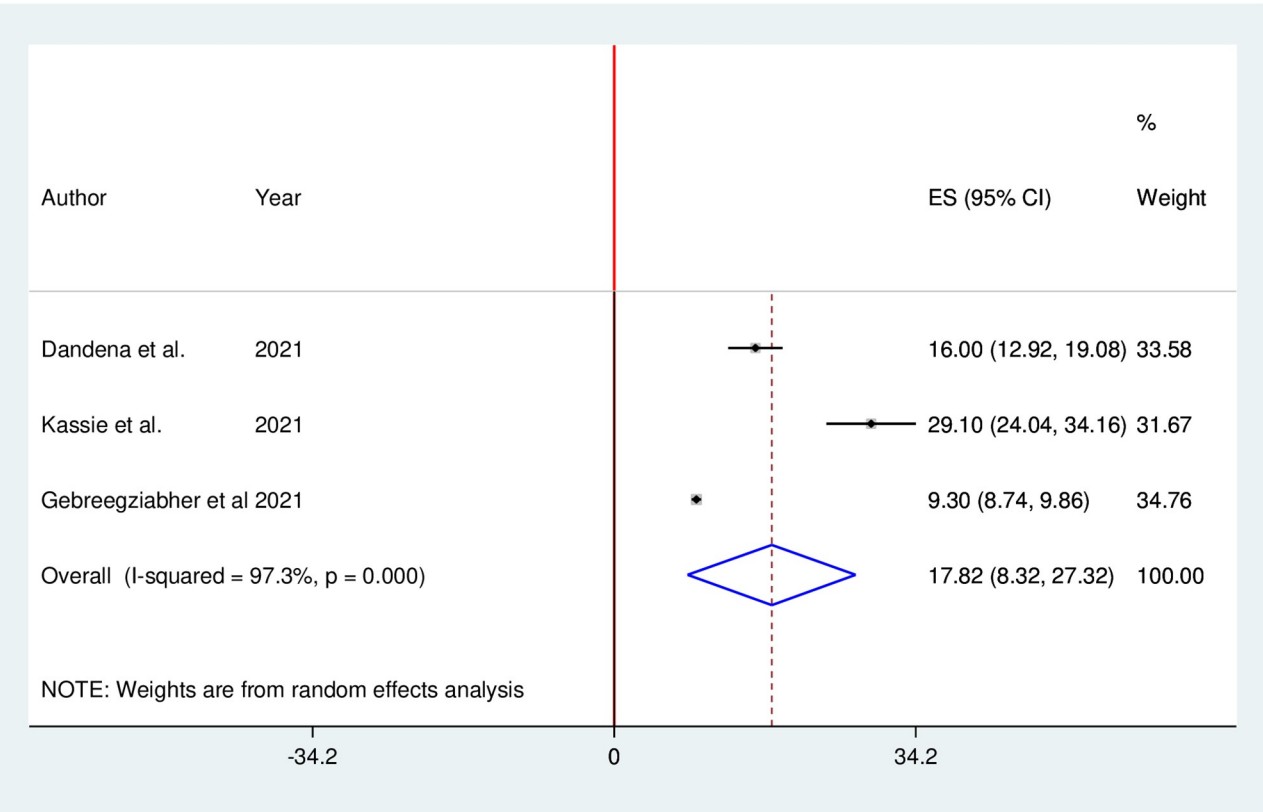

**Fig 11. Forest plot of the pooled reduction of postnatal care services utilization during COVID-19 pandemic in Ethiopia, 2021.**

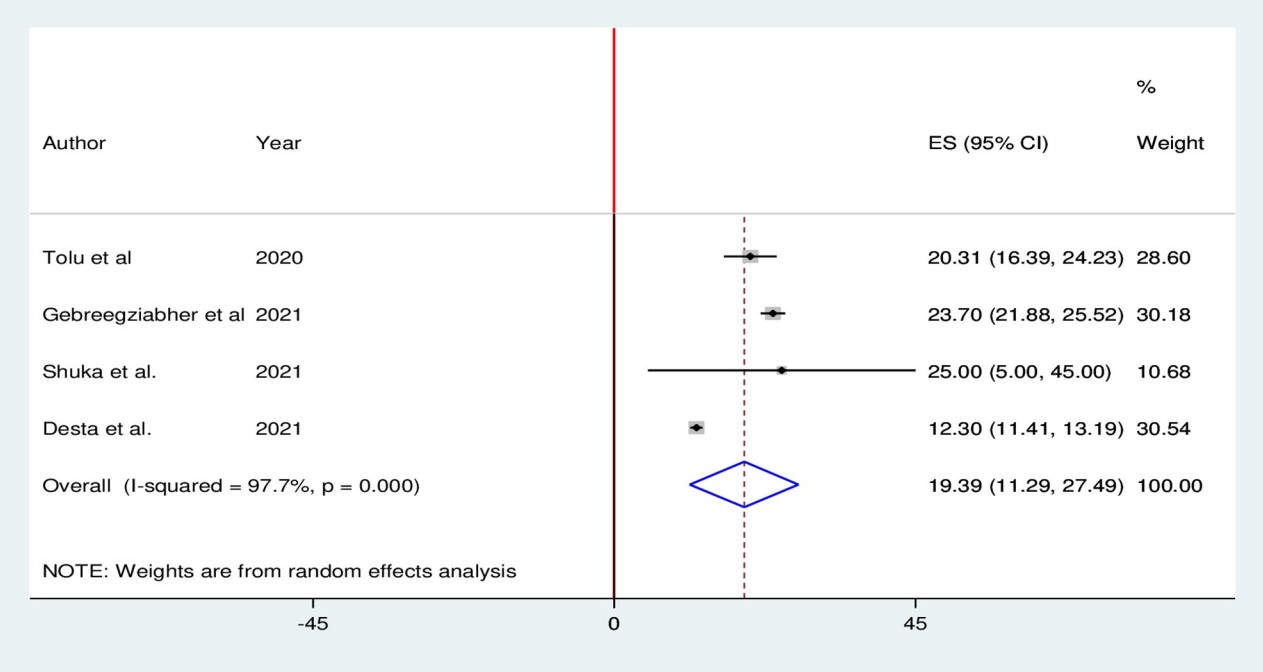

**Fig 12. Forest plot of the pooled reduction of abortion care services utilization during COVID-19 pandemic in Ethiopia, 2021.**

studies done in Belgium [75] and Ireland [76], which reported a substantial reduction in abortion care services during COVID-19. The reason for the reduction in abortion care is that women perhaps engaged less in risky sexual behavior during the COVID-19 pandemic, and were challenged less with unexpected pregnancies [77, 78]. However, evidence also found that demand for self-managed abortions had increased in countries with strict COVID-19 measures [79, 80].

The subgroup analysis of this systematic review and meta-analysis indicated that the highest reduction of family planning, antenatal care, and institutional delivery services was observed in 2020 then services decrease in 2021. This might be due to the fact in the early phase of the pandemic (2020) many health facilities were being used as diagnostic and treatment centers which might directly or indirectly affect essential maternal healthcare service delivery and utilization. The other possible explanation could be due to the effect of media propaganda about the COVID-19 disease which could make the community fear disease transmission and restrict women from using basic healthcare services [81, 82].

This study demonstrated that government measures and health facility barriers such as diversion of resources, staff workload, and burnout, lack of essential drugs, mistreatment and disrespect, weak infrastructure, lack of personal protective equipment and sanitizer affect the provision of maternal healthcare services during the pandemic. This claim is supported by other studies suggesting that the COVID-19 pandemic disrupted essential maternal health services due to resources shifting toward COVID-19 prevention and control activities, and the engagement of staff in COVID-19-related tasks [9, 83–85]. Maternal perception of poor quality of care, lack of transport, cultural events, low economic status, and anxiety and fear of infection, and stigma of COVID-19 were explained as the main barriers that affected essential maternal health services during the pandemic. This finding is substantiated by other findings affirming that maternally related barriers could considerably reduce maternal health services during the pandemic [85–88].

This systematic review and meta-analysis provide vital evidence to inform policy-makers, health programmers, and other relevant stakeholders to take rescale measures for the reduced basic maternal healthcare services utilization during the COVID-19 pandemic. Hereafter, understanding the impact of the COVID-19 pandemic on maternity services utilization could help to design appropriate strategies and interventions for the improvement of maternity services provision and utilization. The main barriers to maternal healthcare services utilization during the COVID-19 pandemic were identified. Hence, prioritizing maternity healthcare provision by considering the challenges should be commenced sooner rather than later.

This review provided comprehensively reviewed the evidence with quantitative pooled reduction of essential maternal healthcare services in Ethiopia. However, this study has the following potential limitations. First, we could not directly compare the utilization of maternal healthcare services before and after the pandemic due to the lack of data on the utilization of maternal health services before the COVID-19 pandemic. Second, this meta-analysis didn't include all regions of the country, which may affect the pooled estimates of maternal healthcare services.

## Conclusion

This study revealed that the utilization of maternal healthcare services in Ethiopia significantly decreased during the COVID-19 pandemic. We identified a significant decrease in the utilization of family planning, antenatal care, institutional delivery, postnatal and abortion care services. This study also demonstrated that maternal perception of poor quality of care and fear of infection, lack of transport, cultural events, diversion of resources, lack of essential drugs,

and lack of personal protective equipment and sanitizer were explained as the main barriers that affect maternal healthcare services during the pandemic. Thus, essential maternity care should be prioritized as an essential core healthcare service by maternal healthcare service providers, policy-makers, programmers, and other relevant stakeholders. Besides, increasing awareness of women through mass media, and making maternity services more accessible and equitable would likely increase the utilization of maternal healthcare services.

## Supporting information

**S1 Table. Preferred Reporting Items for Systematic Reviews and Meta-Analyses (PRISMA) checklist.**
(DOC)

**S2 Table. The Newcastle-Ottawa scale for quality assessment of primary studies.**
(DOCX)

## Acknowledgments

The authors would like to thank the authors of the included primary studies, which were used as a source of information to conduct this systematic review and meta-analysis.

## Author Contributions

**Conceptualization:** Birye Dessalegn Mekonnen.

**Data curation:** Birye Dessalegn Mekonnen.

**Formal analysis:** Birye Dessalegn Mekonnen, Berhanu Wale Yirdaw.

**Investigation:** Birye Dessalegn Mekonnen, Berhanu Wale Yirdaw.

**Methodology:** Berhanu Wale Yirdaw.

**Resources:** Birye Dessalegn Mekonnen.

**Software:** Birye Dessalegn Mekonnen, Berhanu Wale Yirdaw.

**Supervision:** Birye Dessalegn Mekonnen, Berhanu Wale Yirdaw.

**Validation:** Birye Dessalegn Mekonnen.

**Visualization:** Birye Dessalegn Mekonnen, Berhanu Wale Yirdaw.

**Writing – original draft:** Birye Dessalegn Mekonnen, Berhanu Wale Yirdaw.

**Writing – review & editing:** Birye Dessalegn Mekonnen, Berhanu Wale Yirdaw.

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
