## [Decision Letter · Decision Letter 0]

28 Nov 2022

PONE-D-22-03300Impact of COVID-19 Pandemic on Utilization of Essential Maternal Healthcare Services in Ethiopia: A Systematic Review and Meta-AnalysisPLOS ONE

Dear Dr. Mekonnen,

Thank you for submitting your manuscript to PLOS ONE. After careful consideration, we feel that it has merit but does not fully meet PLOS ONE’s publication criteria as it currently stands. Therefore, we invite you to submit a revised version of the manuscript that addresses the points raised during the review process. Please submit your revised manuscript by Dec 10 2022 11:59PM. If you will need more time than this to complete your revisions, please reply to this message or contact the journal office at plosone@plos.org. Please include the following items when submitting your revised manuscript:A rebuttal letter that responds to each point raised by the academic editor and reviewer(s). You should upload this letter as a separate file labeled 'Response to Reviewers'.A marked-up copy of your manuscript that highlights changes made to the original version. You should upload this as a separate file labeled 'Revised Manuscript with Track Changes'.An unmarked version of your revised paper without tracked changes. You should upload this as a separate file labeled 'Manuscript'.

We look forward to receiving your revised manuscript.

Kind regards,

Bidhubhusan Mahapatra, Ph.D.

Academic Editor

PLOS ONE

Journal Requirements:

2**. **Please include captions for your Supporting Information files at the end of your manuscript, and update any in-text citations to match accordingly. Please see our Supporting Information guidelines for more information: http://journals.plos.org/plosone/s/supporting-information.

Additional Editor Comments (if provided):

It is an interesting body of work. I request authors to carefully review the comments provided by the reviewer (specifically reviewer 1) and revise the paper.

Reviewers' comments:

Reviewer's Responses to Questions

**Comments to the Author**

1. Is the manuscript technically sound, and do the data support the conclusions?

Reviewer #1: Partly

Reviewer #2: Yes

Reviewer #3: Yes

2. Has the statistical analysis been performed appropriately and rigorously? 

Reviewer #1: I Don't Know

Reviewer #2: Yes

Reviewer #3: Yes

3. Have the authors made all data underlying the findings in their manuscript fully available?

Reviewer #1: Yes

Reviewer #2: Yes

Reviewer #3: Yes

4. Is the manuscript presented in an intelligible fashion and written in standard English?

Reviewer #1: Yes

Reviewer #2: Yes

Reviewer #3: Yes

5. Review Comments to the Author

Reviewer #1: Thank you for the opportunity to review this paper. The COVID-19 pandemic affected service provision and utilization of several essential services, particularly maternal health. This systematic review focuses on gathering evidence on the effect of the pandemic on service utilization in Ethiopia. It also provides some insights into challenges faced by service users through a qualitative review through meta-analysis. The review also seeks to fill a critical Ethiopia-specific, national-level critical gap and has the potential to make a valuable contribution to the literature. I hope the authors will consider revising the sections below in order to make this paper a strong and insightful contribution. I do not have the expertise to consider the statistics presented in this paper.

The background touches upon multiple issues: the COVID-19 pandemic’s impact on global healthcare, impact on healthcare (and particularly maternal health) services, access to services, the rationale for this study. I would recommend including a section/para on how the pandemic panned out in Ethiopia (when it began, some key policy measures that were taken, etc.) - this would give a reader better context to understand the findings.

There is a section on eligibility criteria in the Methods section. I would recommend restructuring this (maybe using a framework like PICO) to better present the inclusion and exclusion criteria (perhaps even just two sections that say inclusion and exclusion criteria).

In criteria 1, “(1) studies that reported data on the change in the utilization of maternal health services (family planning, antenatal care, institutional delivery, postnatal and abortion care) in Ethiopia were considered;” - please mention the study designs that were included (qualitative or quantitative or mixed methods for example)

In criteria 2, “(2) Both peer reviewed published and unpublished articles were considered;” - please mention what type of unpublished articles were considered. Were these pre-print or reports etc.)?

“A standardized data extraction tool, which was adapted from the Joanna Briggs Institute (JBI) was used to extract data from articles included in the review” - as per my understanding, the JBI tools are used for quality appraisal. Could the authors attach the adapted tool or provide an explanation for why this was used for data extraction instead?

The authors refer to a qualitative review in the results section (abstract and paper both) for the first time. I have two suggestions to the authors: (a) if the aim is to present a meta-analysis and qualitative synthesis of the evidence, please mention this earlier in the methods. For instance, you can say that this is a mixed methods systematic review drawing on mixed methods papers as well in the inclusion criteria. In addition, please mention how you analyzed the qualitative data.; (b) if the aim is to only present a meta-analysis, perhaps it would be more valuable to exclude the qualitative review and present information from such papers in the discussion to support the findings of lower service utilization.

Based on the study characteristics, it appears that 21 studies were selected based on eligibility. Qualitative synthesis was drawn from these 21 studies which seem to include 13 quantitative studies, 5 mixed-methods, 2 pre-post and only 1 qualitative study. Could the authors clarify why and how they did qualitative synthesis of all of these studies, of which 5 were mixed methods and 1 one was qualitative? Were these the 6 excluded from the meta-analysis despite 5 mixed methods studies? A revision of this figure and the first paragraph in study characteristics may be needed as it is a little confusing at present.

In Table 1, Hailemarium et al (2021) is referred to as a qualitative study which reports decreased ANC. Is the table reporting qualitative findings of decreased ANC services or reporting data from a different study/survey?

In the section on Challenges of maternal healthcare services utilization, it is mentioned that 10 studies reported the main challenges that could decrease the utiliization of essential MCH services during the pandemic. Could the authors clarify whether these were qualitative findings from respondents such as pregnant women, policymakers etc.? Or are these proposed challenges? Could the authors also elaborate further on these challenges such as cultural events etc.? It may not be clear to readers on the way some of these challenges manifested.

In the Discussion section, authors refer to “Similar findings have been documented in other studies”. Do they mean other countries or similar settings here?

The authors refer to studies in the UK and Belgium (high income settings with different health systems and pandemic trajectories) when discussing the findings on ANC service utilization. It is unclear what the relevance of these findings would be in this context. Perhaps this could be reframed to make their point clearer.

Could the authors please include a citation for the following sentence: The other possible explanation could be due to the effect of media propaganda about the COVID-19 diseases which could make the community fear disease transmission and restrict women from using basic healthcare services.

The authors could consider adding some information on what the sub-group analysis by publication years revealed about overall service utilization. Perhaps this could be linked - if it makes sense - to policy adaptations.

Minor:

It may be helpful to supply search terms as a supplementary file.

It may also be helpful to include the adapted tool for Newcastle Ottawa as a supplementary file.

Reviewer #2: The review is important in this context and the issue.

Background

It's better to include the local context of maternal health-related information in this section as well. Even if the research was not conducted in these areas, the authors can write how the maternal health services were there before COVID.

Methodology

- Did all the articles include similar family planning devices or different or any family planning devices? It could be included in the criteria.

- Briefly discuss the quality of the articles and also the unpublished materials/reports.

Results

- Check study characteristics line 6 for the review of the sentence.

Discussion

It's better to present the strength of the analysis strongly in the discussion section.

Conclusion

It’s too general recommendation –should work more on it for the conclusive recommendation.

Reviewer #3: Thank you for the opportunity to review this manuscript which reports a meta-analysis and systematic review of research measuring the impact of the COVID-19 pandemic on the utilization of maternal health services in Ethiopia. Remarkably the authors identified 21 papers of which 15 were pooled with meta-analytic techniques. the authors report a 26,6% reduction in maternal health services across a range of activities which is in line with the effects reported in other countries. The articles analyzed were also subject to a qualitative synthesis which shed light on the barriers relevant to seeking maternal health care. these were predictably linked to closures of health care facilities, fear of contracting COVID-19 infection and shortages of personal protective equipment. The service disruptions included gaps in prenatal care, family planning as well as post-natal care.

The authors are to be commended on the robustness of their methods including a comprehensive search strategy and prisma compliant approaches two reporting their findings. The manuscript is well written and clear and presents a thorough analysis and synthesis of the included studies. My primary critique of the paper is the lack of patient reported outcomes in the meta-analysis. While I appreciate that this is outside the scope of the review I think the authors need to make clear that we are not certain that all of the service disruptions were deleterious and lead to harmful outcomes. One could even postulate that there could be opportunities to learn from these reductions with an eye to improving resource stewardship and rationalizing care. If these caveats are addressed, I believe this paper is worthy of publication in the journal.

The results are important in that they confirm that the pattern of service disruptions seen across myriad jurisdictions was not confined too high-income countries and may have even had a higher impact in lower- and middle-income countries.

6. PLOS authors have the option to publish the peer review history of their article (what does this mean?). If published, this will include your full peer review and any attached files.

Reviewer #1: No

Reviewer #2: No

Reviewer #3: No

---

## [Author Response · Author response to Decision Letter 0]

12 Dec 2022

Author’s Point-by-point response letter to reviewer

Title: Impact of COVID-19 Pandemic on Utilization of Essential Maternal Healthcare Services in Ethiopia: A Systematic Review and Meta-Analysis

Author: Birye Dessalegn Mekonnen and Berhanu Wale Yirdaw

Corresponding author: 

Birye Dessalegn Mekonnen 

Email: birye22@gmail.com

Teda Health Science College, Ethiopia 

December 12, 2022 

To Plos One

Dear Sir/Madam, 

This is a point-by-point response letter that accompanies the responses for the editors and reviewers’ comments concerning the manuscript entitled “Impact of COVID-19 Pandemic on Utilization of Essential Maternal Healthcare Services in Ethiopia: A Systematic Review and Meta-Analysis”. It is known that the manuscript has been reviewed by reviewers and sent back to us the authors to carry out the corrections to meet the editor and reviewers’ concern, and for resubmission. 

As authors of this manuscript, the comments and concerns raised by the reviewers’ were highly insightful and enabled us to improve the quality & scientific plausibility of the manuscript. To do so, we have tried to address all the reviewers’ concerns point by point as described below. Therefore, we are pleased to resubmit the revised version of the manuscript for further process and facilitation of its publication on Plos One.

Author’s Point-by-point response to Reviewers’ Comments

Journal Requirements:

1. Please ensure that your manuscript meets PLOS ONE's style requirements, including those for file naming. The PLOS ONE style templates can be found at: https://journals.plos.org/plosone/s/file?id=wjVg/PLOSOne_formatting_sample_main_body.pdf and https://journals.plos.org/plosone/s/file?id=ba62/PLOSOne_formatting_sample_title_authors_affiliations.pdf

Authors’ response: We ensured that our manuscript meets PLOS ONE's style requirements.

Authors’ response: We included captions for the Supporting Information files at the end of the manuscript.

Additional Editor Comments (if provided):

It is an interesting body of work. I request authors to carefully review the comments provided by the reviewer (specifically reviewer 1) and revise the paper.

Authors’ response: Dear editor, first of all thank you very much for your interest for reviewing our paper, and providing constructive and fruitful comments as well as feedback. We carefully reviewed the comments provided by the reviewers, and revised the paper as per your concern. 

Reviewers' comments:

Reviewer's Responses to Questions

Comments to the Author

Reviewer #1: Thank you for the opportunity to review this paper. The COVID-19 pandemic affected service provision and utilization of several essential services, particularly maternal health. This systematic review focuses on gathering evidence on the effect of the pandemic on service utilization in Ethiopia. It also provides some insights into challenges faced by service users through a qualitative review through meta-analysis. The review also seeks to fill a critical Ethiopia-specific, national-level critical gap and has the potential to make a valuable contribution to the literature. I hope the authors will consider revising the sections below in order to make this paper a strong and insightful contribution. I do not have the expertise to consider the statistics presented in this paper.

Authors’ response: Dear reviewer, first of all thank you very much for your interest for reviewing our paper, and providing constructive and fruitful comments as well as feedback. We tried to make revisions on the manuscript as per your comments and concerns. 

The background touches upon multiple issues: the COVID-19 pandemic’s impact on global healthcare, impact on healthcare (and particularly maternal health) services, access to services, the rationale for this study. I would recommend including a section/para on how the pandemic panned out in Ethiopia (when it began, some key policy measures that were taken, etc.) - this would give a reader better context to understand the findings.

Authors’ response: Dear reviewer, again thank you very much. We accepted, and tried to add some information on how the pandemic panned out in Ethiopia as per your recommendation. 

There is a section on eligibility criteria in the Methods section. I would recommend restructuring this (maybe using a framework like PICO) to better present the inclusion and exclusion criteria (perhaps even just two sections that say inclusion and exclusion criteria).

Authors’ response: Dear reviewer, again thank you very much. We accepted, and made revision on the manuscript as per your concern.

In criteria 1, “(1) studies that reported data on the change in the utilization of maternal health services (family planning, antenatal care, institutional delivery, postnatal and abortion care) in Ethiopia were considered;” - please mention the study designs that were included (qualitative or quantitative or mixed methods for example)

Authors’ response: Dear reviewer, again thank you very much. We accepted, and made modification in the manuscript as per your recommendation.

In criteria 2, “(2) Both peer reviewed published and unpublished articles were considered;” - please mention what type of unpublished articles were considered. Were these pres-print or reports etc.)?

Authors’ response: Dear reviewer, again thank you very much. We accepted, and made revision in the manuscript as per your recommendation.

“A standardized data extraction tool, which was adapted from the Joanna Briggs Institute (JBI) was used to extract data from articles included in the review” - as per my understanding, the JBI tools are used for quality appraisal. Could the authors attach the adapted tool or provide an explanation for why this was used for data extraction instead?

Authors’ response: Dear reviewer, again thank you very much. The data have been extracted using a Microsoft Excel sheet. The necessary information (components) from each included article were identified using the Joanna Briggs Institute (JBI); which the JBI tools are also used for data elements collection for systematic reviews in addition to quality appraisal.

The authors refer to a qualitative review in the results section (abstract and paper both) for the first time. I have two suggestions to the authors: (a) if the aim is to present a meta-analysis and qualitative synthesis of the evidence, please mention this earlier in the methods. For instance, you can say that this is a mixed methods systematic review drawing on mixed methods papers as well in the inclusion criteria. In addition, please mention how you analyzed the qualitative data.; (b) if the aim is to only present a meta-analysis, perhaps it would be more valuable to exclude the qualitative review and present information from such papers in the discussion to support the findings of lower service utilization.

Authors’ response: Dear reviewer, again thank you very much. The qualitative data was analyzed based on the main challenges that could decrease the utilization of essential maternal health services during the COVID-19 pandemic. We tried to mention this in the method section, both in the inclusion criteria and data analysis part of the manuscript.

Based on the study characteristics, it appears that 21 studies were selected based on eligibility. Qualitative synthesis was drawn from these 21 studies which seem to include 13 quantitative studies, 5 mixed-methods, 2 pre-post and only 1 qualitative study. Could the authors clarify why and how they did qualitative synthesis of all of these studies, of which 5 were mixed methods and 1 one was qualitative? Were these the 6 excluded from the meta-analysis despite 5 mixed methods studies? A revision of this figure and the first paragraph in study characteristics may be needed as it is a little confusing at present.

Authors’ response: Dear reviewer, again thank you very much. We tried revise the figure. We reviewed the challenges from both the qualitative and quantitative studies.

In Table 1, Hailemarium et al (2021) is referred to as a qualitative study which reports decreased ANC. Is the table reporting qualitative findings of decreased ANC services or reporting data from a different study/survey?

Authors’ response: Dear reviewer, again thank you very much. Yes, the table reported qualitative findings of decreased ANC services. We reported this finding as this primary study documented that ANC service was decreased during Covid-19.

In the section on Challenges of maternal healthcare services utilization, it is mentioned that 10 studies reported the main challenges that could decrease the utiliization of essential MCH services during the pandemic. Could the authors clarify whether these were qualitative findings from respondents such as pregnant women, policymakers etc.? Or are these proposed challenges? Could the authors also elaborate further on these challenges such as cultural events etc.? It may not be clear to readers on the way some of these challenges manifested.

Authors’ response: Dear reviewer, again thank you very much. The challenges stated in the manuscript are the finding of in the primary studies; they are not proposed challenges. The studies reported the main challenges that could decrease the utilization of essential MCH services during the pandemic include: qualitative study, mixed (qualitative and quantitative) methods, and some are quantitative which assessed the main challenges quantitatively. Hence, it is difficult to elaborate further on these challenges such as cultural events because we stated all the identified challenges in the primary studies.

In the Discussion section, authors refer to “Similar findings have been documented in other studies”. Do they mean other countries or similar settings here?

Authors’ response: Dear reviewer, again thank you very much. Her we mean “Similar findings have been documented in other countries”. We made correction in the manuscript as per your concern.

The authors refer to studies in the UK and Belgium (high income settings with different health systems and pandemic trajectories) when discussing the findings on ANC service utilization. It is unclear what the relevance of these findings would be in this context. Perhaps this could be reframed to make their point clearer.

Authors’ response: Dear reviewer, again thank you very much. As per our search, we couldn’t find other similar studies in similar settings that could show the impact of Covid-19 on ANC. In addition, we used this comparison to discuss whether the pandemic affects both high- and low-income countries (its global impact on essential maternal health services with appreciating the setting difference). Thus, we the authors agreed to leave (not change) this part in the discussion as it is, after through discussion.

Could the authors please include a citation for the following sentence: The other possible explanation could be due to the effect of media propaganda about the COVID-19 diseases which could make the community fear disease transmission and restrict women from using basic healthcare services.

Authors’ response: Dear reviewer, again thank you very much. We included citations for the above-mentioned statement in the manuscript as per your suggestion. 

The authors could consider adding some information on what the sub-group analysis by publication years revealed about overall service utilization. Perhaps this could be linked - if it makes sense - to policy adaptations.

Authors’ response: Dear reviewer, again thank you very much. We already tried to provide some information on what the sub-group analysis by publication years revealed.

Minor:

It may be helpful to supply search terms as a supplementary file. It may also be helpful to include the adapted tool for Newcastle Ottawa as a supplementary file.

Authors’ response: Dear reviewer, again thank you very much. We included supplementary files for both search terms and Newcastle Ottawa.

Reviewer #2: The review is important in this context and the issue.

Background

It's better to include the local context of maternal health-related information in this section as well. Even if the research was not conducted in these areas, the authors can write how the maternal health services were there before COVID.

Authors’ response: Dear reviewer, first of all thank you very much for your interest for reviewing our work, and providing constructive and fruitful comments. It is difficult to write how the maternal health services were there before COVID because we didn’t review on this regard. Also, we mentioned it as limitation of the review.

Methodology

- Did all the articles include similar family planning devices or different or any family planning devices? It could be included in the criteria. 

Authors’ response: Dear reviewer, again thank you very much. This review considered ‘family planning utilization’ with any family planning devices.

- Briefly discuss the quality of the articles and also the unpublished materials/reports.

Authors’ response: Dear reviewer, again thank you very much. We included the quality assessment for each article as supplementary.

Results

- Check study characteristics line 6 for the review of the sentence.

Authors’ response: Dear reviewer, again thank you very much. We reviewed and made some correction.

Discussion

It's better to present the strength of the analysis strongly in the discussion section.

Authors’ response: Dear reviewer, again thank you very much. We tried to present the strength of the analysis or review in the discussion section as per your suggestion.

Conclusion

It’s too general recommendation –should work more on it for the conclusive recommendation.

Authors’ response: Dear reviewer, again thank you very much. We made some revision on this regard.

Reviewer #3:

Thank you for the opportunity to review this manuscript which reports a meta-analysis and systematic review of research measuring the impact of the COVID-19 pandemic on the utilization of maternal health services in Ethiopia. Remarkably the authors identified 21 papers of which 15 were pooled with meta-analytic techniques. the authors report a 26,6% reduction in maternal health services across a range of activities which is in line with the effects reported in other countries. The articles analyzed were also subject to a qualitative synthesis which shed light on the barriers relevant to seeking maternal health care. these were predictably linked to closures of health care facilities, fear of contracting COVID-19 infection and shortages of personal protective equipment. The service disruptions included gaps in prenatal care, family planning as well as post-natal care.

The authors are to be commended on the robustness of their methods including a comprehensive search strategy and prisma compliant approaches two reporting their findings. The manuscript is well written and clear and presents a thorough analysis and synthesis of the included studies. My primary critique of the paper is the lack of patient reported outcomes in the meta-analysis. While I appreciate that this is outside the scope of the review, I think the authors need to make clear that we are not certain that all of the service disruptions were deleterious and lead to harmful outcomes. One could even postulate that there could be opportunities to learn from these reductions with an eye to improving resource stewardship and rationalizing care. If these caveats are addressed, I believe this paper is worthy of publication in the journal.

The results are important in that they confirm that the pattern of service disruptions seen across myriad jurisdictions was not confined too high-income countries and may have even had a higher impact in lower- and middle-income countries.

Authors’ response: Dear reviewer, again thank you very much for your constructive and fruitful feedback. We appreciated your insightful observations. We couldn’t address your concerns as patient reported outcomes are not the objective of this systematic review and meta-analysis. Finally, we really would like to thank you for your constructive feedbacks, and devoted time to review our work. 

.

I look forward to hearing from you at your earliest convenience.

Birye Dessalegn Mekonnen (MPH in Reproductive and Child Health)

Corresponding author

---

## [Decision Letter · Decision Letter 1]

10 Jan 2023

PONE-D-22-03300R1Impact of COVID-19 Pandemic on Utilization of Essential Maternal Healthcare Services in Ethiopia: A Systematic Review and Meta-AnalysisPLOS ONE

Dear Dr. Mekonnen,

Thank you for submitting your manuscript to PLOS ONE. After careful consideration, we feel that it has merit but does not fully meet PLOS ONE’s publication criteria as it currently stands. Therefore, we invite you to submit a revised version of the manuscript that addresses the points raised during the review process.

We look forward to receiving your revised manuscript.

Kind regards,

Bidhubhusan Mahapatra, Ph.D.

Academic Editor

PLOS ONE

Journal Requirements:

Additional Editor Comments (if provided):

The paper requires a thorough proofread and edits.

Reviewers' comments:

Reviewer's Responses to Questions

**Comments to the Author**

1. If the authors have adequately addressed your comments raised in a previous round of review and you feel that this manuscript is now acceptable for publication, you may indicate that here to bypass the “Comments to the Author” section, enter your conflict of interest statement in the “Confidential to Editor” section, and submit your "Accept" recommendation.

Reviewer #1: All comments have been addressed

Reviewer #2: All comments have been addressed

2. Is the manuscript technically sound, and do the data support the conclusions?

Reviewer #1: Yes

Reviewer #2: Yes

3. Has the statistical analysis been performed appropriately and rigorously? 

Reviewer #1: I Don't Know

Reviewer #2: Yes

4. Have the authors made all data underlying the findings in their manuscript fully available?

Reviewer #1: Yes

Reviewer #2: Yes

5. Is the manuscript presented in an intelligible fashion and written in standard English?

Reviewer #1: (No Response)

Reviewer #2: Yes

6. Review Comments to the Author

Reviewer #1: Dear authors,

Thank you for addressing comments point by point and revising the paper where you agree. I have a couple of minor suggestions - please include information on when the first COVID-19 case was identified in Ethiopia when you speak of the measures put in place.

Please also proofread for language or typing errors (eg: PICO not PICOS framework).

Reviewer #2: Congratulations. The authors have responded all the comments. Please check the formatting and minor grammatical errors (for example - two full stop).

7. PLOS authors have the option to publish the peer review history of their article (what does this mean?). If published, this will include your full peer review and any attached files.

Reviewer #1: No

Reviewer #2: No

---

## [Author Response · Author response to Decision Letter 1]

15 Jan 2023

Author’s Point-by-point response letter to the reviewers

Title: Impact of COVID-19 Pandemic on Utilization of Essential Maternal Healthcare Services in Ethiopia: A Systematic Review and Meta-Analysis

Author: Birye Dessalegn Mekonnen and Berhanu Wale Yirdaw

Corresponding author: 

Birye Dessalegn Mekonnen 

Email: birye22@gmail.com

Teda Health Science College, Ethiopia 

January 15, 2023 

To Plos One

Dear Sir/Madam, 

This is a point-by-point response letter that accompanies the responses to the editor’s and reviewers’ comments concerning the manuscript entitled “Impact of COVID-19 Pandemic on Utilization of Essential Maternal Healthcare Services in Ethiopia: A Systematic Review and Meta-Analysis”. It is known that the manuscript has been reviewed by reviewers and sent back to us by the authors to carry out the corrections to meet the editor and reviewers’ concerns, and for resubmission. 

As authors of this manuscript, the comments and concerns raised by the reviewers were highly insightful and enabled us to improve the quality & scientific plausibility of the manuscript. To do so, we have tried to address all the reviewers’ concerns point by point as described below. Therefore, we are pleased to resubmit the revised version of the manuscript for further process and facilitation of its publication on Plos One.

Author’s Point-by-point response to Reviewers’ Comments

Journal Requirements:

Authors’ response: We carefully reviewed and ensured that the reference list is complete and correct. Also, we didn’t cite a retracted article.

Additional Editor Comments (if provided):

The paper requires a thorough proofread and edits.

Authors’ response: Dear editor, first of all, thank you very much for providing us second opportunity to revise or amend our paper. We made language edits and corrections on some typing errors after a thorough review of the paper. We really would like to thank you for your patience, and devoted time to manage our paper.

Reviewers' comments:

Reviewer #1: Dear authors,

Thank you for addressing comments point by point and revising the paper where you agree. I have a couple of minor suggestions - please include information on when the first COVID-19 case was identified in Ethiopia when you speak of the measures put in place.

Please also proofread for language or typing errors (eg: PICO not PICOS framework).

Authors’ response: Dear reviewer, first of all, thank you very much for your interest and devoted time to reviewing our work for the second time. We accepted your suggestion, and include information on when the first COVID-19 case was identified in Ethiopia (i.e March 13, 2020) in the manuscript. Additionally, we made language edits and corrections on some typing errors after a thorough review of the paper as per your suggestion.

Reviewer #2: Congratulations. The authors have responded all the comments. Please check the formatting and minor grammatical errors (for example - two full stop).

Authors’ response: Dear reviewer, first of all, thank you very much for your interest and devoted time to reviewing our work for the second time. We made language edits and corrections on some grammatical errors after a thorough review of the paper.

I look forward to hearing from you at your earliest convenience.

Birye Dessalegn Mekonnen (MPH in Reproductive and Child Health)

Corresponding author

---

## [Editor Report · Decision Letter 2]

19 Jan 2023

Impact of COVID-19 Pandemic on Utilization of Essential Maternal Healthcare Services in Ethiopia: A Systematic Review and Meta-Analysis

PONE-D-22-03300R2

Dear Dr. Mekonnen,

We’re pleased to inform you that your manuscript has been judged scientifically suitable for publication and will be formally accepted for publication once it meets all outstanding technical requirements.

Kind regards,

Bidhubhusan Mahapatra, Ph.D.

Academic Editor

PLOS ONE